# Understanding the psychological impact of the COVID-19 pandemic and containment measures: An empirical model of stress

**Bartholomäus Wissmath**[1,2,3]*, **Fred W. Mast**[1,3], **Fabian Kraus**[1,2], **David Weibel**[1,2,3]

**1** Institute of Psychology, University of Bern, Bern, Switzerland, **2** w hoch 2 GmbH, Bern, Switzerland, **3** Swiss Distance Learning University, Brig, Switzerland

* wissmath@psy.unibe.ch

**Data Availability Statement:** Understanding the psychological impact of the COVID-19 pandemic and containment measures: an empirical model of

## Abstract

Epidemics such as COVID-19 and corresponding containment measures are assumed to cause psychological stress. In a survey during the lockdown in Switzerland (n = 1565), we found substantially increased levels of stress in the population. In particular, individuals who did not agree with the containment measures, as well as those who saw nothing positive in the crisis, experienced high levels of stress. In contrast, individuals who are part of a risk group or who are working in healthcare or in essential shops experienced similar stress levels as the general public. The psychological mechanisms that determine stress, caused by the COVID-19 pandemic and containment measures, are not yet clear. Thus, we conducted a path analysis to gain a deeper understanding of the psychological mechanisms that lead to stress. Experiencing fear of the disease is a key driver for being worried. Our model further shows that worries about the individual, social, and economic consequences of the crisis, strongly boost stress. The infection rate in the canton (i.e., state) of residence also contributes to stress. Positive thinking and perceived social, organizational, and governmental support mitigate worries and stress. Our findings indicate that containment measures increase worries and stress, especially for those who feel that these measures either are not sufficient or go too far. Thus, highlighting positive aspects of the crisis and convincing people of the effectiveness and necessity of mitigation measures can, not only promote compliance, but also reduce stress. Our model suggests that people who feel protected by the authorities have fewer worries, which can, in turn, limit the negative impact of the crisis on mental health.

## Introduction

COVID-19 evokes stress in patients, healthcare professionals, and relatives [1]. The impact may not be limited to those directly affected; health worries and uncertainty are assumed to generate fear, anxiety, and severe stress in the general population [2]. Hence, mental health practitioners anticipate a sharp rise in the need for mental health services [3]. Some individuals may be more vulnerable to psychosocial consequences; risk factors include certain personality

stress [Dataset]. Available from: https://osf.io/grvwa/. Deposited May 13, 2020.

**Funding:** The authors received no specific funding for this work.

**Competing interests:** The authors have declared that no competing interests exist.

traits (e.g., neuroticism, alexithymia), mental or physical pre-existing conditions, as well as previous traumatic experiences (e.g., childhood maltreatment), or the individual's current situation, such as working in healthcare, social isolation, or poverty [4–10].

Switzerland was among the countries first and most affected by COVID-19 in Europe. To curb the spread of the virus, the Swiss government enacted a set of containment measures: encouraging hygiene, giving shelter-in-place orders, as well as closing borders, child-care facilities, schools, restaurants, bars, leisure facilities, and non-essential shops. Events and gatherings of more than five individuals were banned.

These measures directly affect the general population and may come with side effects. Reduced social interactions are risk factors for mental disorders such as major depression [11], and the containment measures may trigger an economic downturn, thus adding further stress [12]. Early studies on COVID-19 observed increased stress levels [13]. However, little is known about the psychological mechanisms that determine individual stress in this crisis. It is unclear which individuals are particularly affected by stress, which factors have an increasing or decreasing influence on stress, and how these factors interact. In particular, the interplay between fear of the virus, infection rate, risk factors, individual and collective resources, perception of the containment strategy, and psychological stress has not yet been fully understood. The aim of the present study is to address these questions by using path analysis. We assume that fear of COVID-19 and the local infection rate trigger stress [14, 15]. Individual, economic, and societal worries are also expected to increase stress levels. Agreement with containment measures, perceived support, [14, 16], and being optimistic about the crisis [17], are likely to decrease stress. During the lockdown, we surveyed 1565 individuals in Switzerland online.

## Materials and methods

### Participants

From March 27 to April 26, 2020, a total of 1565 individuals completed the online survey. Participants were recruited via academic institutions, Facebook, and LinkedIn. Posts on social media platforms were published on March 31, 2020 (German) and April 1 (French and Italian). Posts on Facebook were boosted by paid promotion, thereby reaching 1515 (German), 420 (French) and 1006 (Italian) post engagements. Before starting the survey, participants were provided with information about the purpose of the study, data protection policy, and the institution and authors responsible for the study. Individuals from German-speaking (n = 1206), French-speaking (n = 247), and Italian-speaking (n = 112) regions were surveyed. A total of 215 individuals had not completed or had cancelled the survey. The data of three participants were removed, since they had requested the deletion of their data after completion of the survey. An examination of the data for repeated or non-human responses did not reveal any indication of such responses. According to their demographic information, all participants resided in a Swiss canton (as opposed to living abroad). Further information about the sample is provided in Table 1. The study was reviewed and approved by the Ethics Committee of the Faculty of Human Sciences, University of Bern, Switzerland.

### Measurements

**Stress.** We used the German [18], French [19], and Italian [20] version of the Perceived Stress Scale (PPS-10) (10-items) [21, 22]. The widely used PSS is based on the stress-related components of unpredictable, uncontrollable, and overloading life events (e.g., "In the last month, how often have you been angry because of things that were outside your control?"), (0 = never, 1 = almost never, 2 = sometimes, 3 = fairly often, 4 = very often). A PSS score was calculated for each participant by summing up across all scale items. Reliability, calculated for

**Table 1. Group comparisons of experienced stress.**

| Variable | Sample statistics (N = 1565) | | | | ANOVA statistics | | |
|---|---|---|---|---|---|---|---|
| | n | Percentage (%) | Mean PSS Score | SD | F | p | Effect size |
| **Sex[+]** | | | | | 12.6 | < .001* | $\eta^2 = 0.016$ |
| (A) Female | 1043 | 66.6 | 16.1 | 6.24 | | | |
| (B) Male | 516 | 33.0 | 14.4 | 6.84 | | | |
| (C) Other / not specified | 6 | 0.4 | 17.5 | 10.4 | | | |
| **Age[+]** | | | | | 8.84 | < .001* | $\eta^2 = 0.028$ |
| (A) <25 | 126 | 8.1 | 17.4 | 6.87 | | | |
| (B) 25–34 | 371 | 23.7 | 16.1 | 6.10 | | | |
| (C) 35–44 | 380 | 24.3 | 16.2 | 6.29 | | | |
| (D) 45–54 | 295 | 18.8 | 14.9 | 6.78 | | | |
| (E) 55–64 | 261 | 16.7 | 15.1 | 6.67 | | | |
| (F) > 64 | 132 | 8.4 | 12.9 | 5.95 | | | |
| **Education** | | | | | 2.66 | 0.014 | $\eta^2 = 0.01$ |
| (A) no finished education | 5 | 0.3 | 13.2 | 4.02 | | | |
| (B) mandatory school time (9 years) | 40 | 2.6 | 15.8 | 6.77 | | | |
| (C) apprenticeship, vocational school, commercial school | 311 | 19.9 | 15.3 | 6.97 | | | |
| (D) High School | 99 | 6.3 | 17.6 | 6.9 | | | |
| (E) Higher technical or vocational school | 298 | 19.0 | 14.8 | 6.28 | | | |
| (F) University | 778 | 49.7 | 15.6 | 6.28 | | | |
| (G) Other | 34 | 34 | 16.7 | 6.94 | | | |
| **Type of household** | | | | | 2.19 | 0.042 | $\eta^2 = 0.008$ |
| (A) Single person household | 327 | 20.9 | 15.5 | 7.14 | | | |
| (B) Non-family household with multiple persons | 127 | 8.1 | 15.8 | 6.05 | | | |
| (C) Couple without kids in the household | 440 | 28.1 | 14.8 | 6.29 | | | |
| (D) Couple with kids in the household | 469 | 30.0 | 15.8 | 6.14 | | | |
| (E) Single parent with kids in the household | 143 | 9.1 | 16.7 | 6.86 | | | |
| (F) Multiple-family household | 58 | 3.7 | 15.4 | 6.93 | | | |
| (G) Retirement or nursing home | 1 | 0.1 | 25.0 | N/A (n = 1) | | | |
| **Having Kids** | | | | | 1.58 | 0.209 | d = 0.077 |
| Yes | 472 | 30.2 | 15.9 | 6.33 | | | |
| No | 1093 | 69.8 | 15.4 | 6.58 | | | |
| **Employment at the beginning of crisis** | | | | | 0.59 | 0.442 | d = -0.046 |
| Yes | 1252 | 80.0 | 15.5 | 6.31 | | | |
| No | 313 | 20.0 | 15.8 | 7.24 | | | |
| **Working in healthcare[1)]** | | | | | 0.42 | 0.516 | d = 0.048 |
| Yes | 235 | 15.0 | 15.7 | 5.82 | | | |
| No | 1017 | 65.0 | 15.4 | 6.42 | | | |
| **Working in an essential shop[1)]** | | | | | 0.83 | 0.361 | d = 0.111 |
| Yes | 91 | 5.8 | 16.1 | 5.57 | | | |
| No | 1161 | 74.2 | 15.4 | 6.36 | | | |
| **Part of a risk group[2)]** | | | | | 4.77 | 0.009 | $\eta^2 = 0.006$ |
| (A) Yes | 265 | 16.9 | 15.4 | 7.27 | | | |
| (B) No | 1180 | 75.4 | 15.4 | 6.30 | | | |
| (C) Don't know | 120 | 7.7 | 17.3 | 6.52 | | | |
| **Agreement with government's containment strategy[+]** | | | | | 48.8 | < .001* | $\eta^2 = 0.036$ |
| (A) Agreement | 1036 | 66.2 | 14.8 | 6.10 | | | |
| (B) No agreement | 425 | 27.2 | 16.7 | 6.46 | | | |

*(Continued)*

**Table 1.** (Continued)

| Variable | Sample statistics (N = 1565) | | | | ANOVA statistics | | |
|---|---|---|---|---|---|---|---|
| | n | Percentage (%) | Mean PSS Score | SD | F | p | Effect size |
| (C) No agreement at all | 104 | 6.6 | 18.8 | 8.55 | | | |
| **Following shelter-in-place orders stringently** | | | | | 3.19 | 0.074 | d = 0.108 |
| Yes | 1041 | 66.5 | 15.8 | 6.53 | | | |
| No | 524 | 33.5 | 15.1 | 6.44 | | | |
| **News consumption** | | | | | 3.23 | 0.012 | $\eta^2$ = 0.008 |
| (A) Never | 15 | 1.0 | 17.1 | 8.20 | | | |
| (B) 1–2 times per week | 126 | 8.1 | 14.6 | 6.53 | | | |
| (C) 3–4 times per week | 157 | 10.0 | 15.5 | 5.81 | | | |
| (D) Once per day | 634 | 40.5 | 15.1 | 6.21 | | | |
| (E) Multiple times per day | 633 | 40.4 | 16.2 | 6.86 | | | |
| **Estimated duration of the crisis** | | | | | 0.90 | 0.496 | $\eta^2$ = 0.003 |
| (A) 2 weeks | 30 | 1.9 | 14.8 | 7.40 | | | |
| (B) 1 month | 163 | 10.4 | 14.8 | 6.32 | | | |
| (C) 2 months | 555 | 35.5 | 15.6 | 6.47 | | | |
| (D) 3–5 months | 484 | 30.9 | 15.8 | 6.20 | | | |
| (E) 6 months | 161 | 10.3 | 15.1 | 6.45 | | | |
| (F) 1 year | 74 | 4.7 | 15.8 | 7.23 | | | |
| (G) longer than a year | 98 | 6.3 | 16.2 | 7.60 | | | |
| **Seeing positive aspects of the crisis** | | | | | 87.6 | < .001* | d = 0.767 |
| Yes | 1415 | 90.4 | 15.1 | 6.15 | | | |
| No | 150 | 9.6 | 20.2 | 7.85 | | | |

PSS: Perceived Stress Scale; SD: Standard Deviation; p = significance of ANOVA main effects;

+ = significant post hoc Bonferroni tests (p < .001): Sex: (A) > (B), Age: (A) > (F), Agreement with containment strategy: (A) < (B) < (C);

* = significant after Bonferroni correction ($\alpha$ = 0.05/14 = 0.00357);

[1] unemployed participants were excluded from this analysis.

[2] at the time defined by the Swiss authorities as being either > 65 years old or being adult and having high blood pressure, diabetes, cardiovascular disease, respiratory disease or cancer.

all participants and all versions of the scale (German, French, and Italian), was high (Cronbach's alpha = .86).

**Amount of worries.** We assessed worries about the following aspects (0 = not at all, 1 = little, 2 = medium, 3 = strongly, 4 = very strongly): physical health, mental health, health of family and friends, personal safety, social life, private life and personal needs, financial situation, job security, economic situation, healthcare, and basic supply. A principal component analysis on these items suggests a one-factorial solution. We computed the mean value across all items.

**Fear of COVID-19.** Fear was measured with a single item: "How afraid are you of the coronavirus (COVID-19)?" (0 = not at all, 1 = little, 2 = medium, 3 = strongly, 4 = very strongly).

**Support.** Participants assessed perceived support from the following: family, friends and social network, neighbors, employer, authorities, day-care centers, schools, church or religious community, primary care physician, and hospitals (0 = not at all, 1 = little, 2 = partially, 3 = strongly, 4 = very strongly). For the subsequent analysis, a perceived support score was calculated for each participant by summing up all items.

**Infection rate.** The cases per 100,000 inhabitants in the participants' resident canton (state) were used for the path model [23].

**Agreement with containment measures.** We asked the participants whether they thought that the measures were (a) not at all strict enough (b) not strict enough (c) just right (d) too strict (e) much too strict. For the analyses, the categories (a) and (e) were combined as "no agreement at all," and the categories (b) and (d) were combined as "no agreement." These combined categories were contrasted with (c) "agreement."

**Following shelter-in-place orders stringently.** We asked the participants if they followed the shelter-in-place orders stringently (1 = No; 2 = Yes).

**Seeing positive aspects.** We asked the respondents whether there were any positive aspects for them in the current situation (1 = No; 2 = Yes).

**Participants' individual situation and demographics.** We assessed sex, age, education, type of household, having kids, employment, working in healthcare or essential shops with customer interaction, part of a risk group (at the time, defined as >64 years old or adult with high blood pressure, diabetes, cardiovascular disease, respiratory disease, or cancer), media consumption, and participants' estimation of the length of the crisis.

## Statistical analysis

In a first step, a one-sample z-test was conducted to compare the stress level of the study sample to a representative community sample under normal conditions [22]. Before the test was carried out, we checked whether the stress values were normally distributed: the stress scores were approximately normally distributed, as the skewness equaled 0.43 (a normal distribution can be assumed for skewness values between -0.5 and 0.5 [24]). Thus, the data allowed for z-test as well as ANOVAs (step 2 below).

In a second step, an ANOVA was carried out to compare different subgroups of the study sample. Because of multiple comparisons, Bonferroni's adjustments were made to prevent Type I error inflation ($\alpha = 0.05/14 = 0.00357$). IBM SPSS Statistics for Windows, Version 25.0 was used to run this analysis [25].

In a third step, we tested how stress is related to various variables that are assumed to affect well-being in the context of the pandemic. A path analysis was conducted with stress as the target variable. The path analysis was carried out with IBM SPSS Amos, Version 25.0 [26], using the maximum likelihood method. To test for normality prior to the analysis, a descriptive approach was used. In the context of SEM or path analysis, kurtosis values of $> 7$ indicate a substantial deviation from normality [27, 28]. In terms of skewness values of $> 3$ indicate extreme levels of skewness [29]. For both skewness and kurtosis, our values were below these thresholds and thus, the use of the maximum likelihood method was appropriate. To test the significance of the path coefficients, t-tests were calculated. The following variables were included in the path model: stress, fear of COVID-19, worries, support, and infection rate. These variables were selected because previous findings suggest that they affect stress. In addition, "agreement with containment measures" as well as "positive aspects" were included, because the second step of analysis (see above) showed that these variables exert the strongest influence on stress.

We used $p < 0.05$ as a priori level of significance. However, we reported lower p-values when appropriate (e.g., $p < 0.001$). Effect sizes were reported as small, medium, or strong according to the heuristics of Eid, Gollwitzer and Schmitt [30].

## Results

First, we compared the observed stress levels in our sample ($M = 15.58$; $SD = 6.65$) to a representative community sample under normal conditions ($M = 12.57$; $SD = 6.24$) [22]. A one-

sample z-test showed higher stress levels during the lockdown period ($z = 19.08$, p < .001). The effect (d = .48) was medium to strong.

In a second step, we computed multiple comparisons (ANOVA) of different subgroups (see Table 1). Women expressed higher stress levels than men ($p < .001$), as has been found in non-epidemic situations [21]. Young individuals reported the highest stress levels, whereas the levels were lowest for individuals older than 65 years ($p < .001$; small to medium effect); the other age groups did not differ significantly. The majority of respondents (66%) agreed with the measures of the Swiss authorities. These individuals scored lower on stress than individuals who felt that the measures were either not sufficient or too extreme ($p < .001$). Individuals who strongly disagreed experienced even more stress (medium effect). Individuals who reported that the lockdown situation also has positive aspects (89.4%) had substantially lower stress levels compared to those who did not see any positive aspects at all ($p < .001$; strong effect).

No other assessed variable influenced stress (cf. Table 1). Interestingly, individuals identifying as members of a risk group ($p = 0.009$; non-significant after Bonferroni-correction; $\alpha = 0.05/14 = 0.00357$), as well as individuals working in healthcare ($p = 0.516$) or in essential shops with customer contact ($p = 0.361$), did not experience more stress than the rest of the population. Also, whether or not participants followed the shelter-in-place orders stringently was not related to stress ($p = .074$). The expected duration of the pandemic (p = 0.496) and amount of news consumption ($p = 0.012$, non-significant after Bonferroni-correction; see above) had no influence on stress.

In a third step, we conducted a path analysis with stress as the target variable. Fig 1 depicts the resulting model, showing a good fit.

In total, the measured variables explain 30% of variance in the endogenous variable, stress. Worries contribute the most to stress, and worries mediate between fear of COVID-19 and stress. Seeing positive aspects of the crisis, as well as agreement with the government's containment strategy, are correlated, and both mitigate worries and boost perceived support. In addition, positive thinking directly minimizes stress. Perceived support from family, friends, organizations, and authorities mitigates worries and reduces stress.

## Discussion

Our results show that the COVID-19 pandemic and the mitigation strategies increase the population's stress level. We present a model that builds upon existing theories of stress and explains 30% of the variance in stress.

In our model, fear of the virus is the most important booster of worries and the amount of worries is the most important driver of stress. Agreeing with the authorities' containment strategy and seeing positive aspects of the crisis are important factors mitigating stress. Correspondingly, feeling that the containment measures are not sufficient or too extreme is associated with more stress. Support provided by family, friends, organizations, and authorities are also important protective factors against stress.

In our study, risk factors for high stress levels are a high amount of worries, perceived lack of support and, to a lesser extent, high infection rates. Eleven percent of the respondents did not see anything positive about the crisis. This group experienced substantially more stress than individuals who saw at least one positive aspect. Although seeing something positive about the crisis may be associated with the individual situation, personality traits, and respondent's coping behavior [17], our findings indicate that recognizing positive aspects alleviates stress. Some individuals, however, did not see anything positive about the crisis. This raises the question as to why their assessment of the situation is so negative. For example, childhood

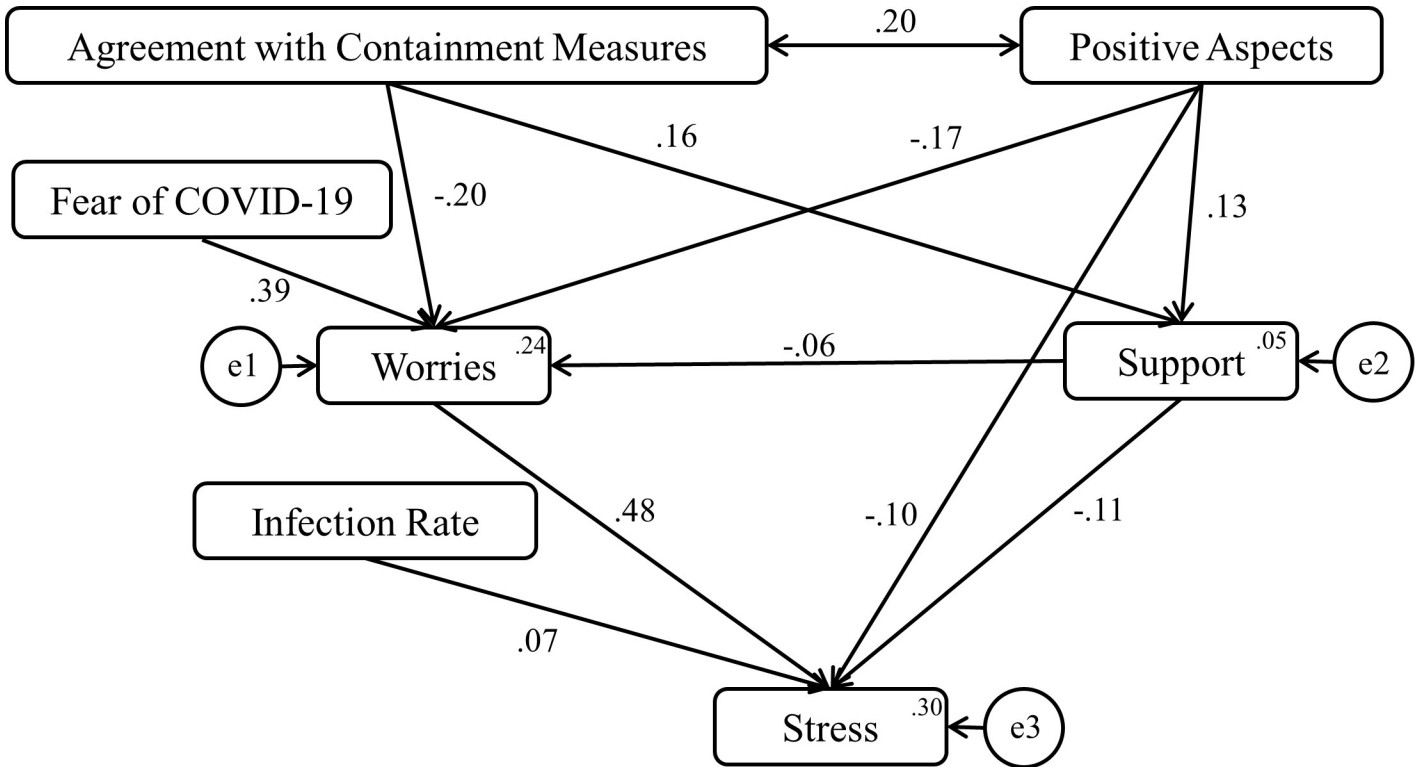

**Fig 1. A path model showing the influence of various determinants on stress.** Model fit statistics: $\chi^2$ (10) = 42.85, p < .01, GFI = .99, CFI = .97, NFI = .96, TLI = .94, RMSEA = .05. All path coefficients turned out to be highly significant (all paths p < .001, except path between Support and Worries: p < .005). Error variances appear in small circles.

maltreatment is associated with hopelessness [31] and psychological distress [5]. A recent study in connection with COVID-19 indicates that pre-pandemic maltreatment makes people particularly vulnerable to negative psychosocial consequences of the pandemic [32]. In addition, other factors and traits–such as pessimistic thinking [e.g., 33] or generalized anxiety [e.g., 34]–could account for the lack of perspective. Surprisingly, members of the risk groups, old respondents, and individuals working in healthcare or essential shops with customer contact, did not score higher on stress. One reason could be that extensive protection measures had been mandated. In addition, healthcare providers in Switzerland have, so far, not been overwhelmed by cases of COVID-19. In contrast to agreement with the government's containment strategies, there was no association between following the shelter-in-place orders stringently and perceived stress. About a third of the respondents had indicated that they did not follow the orders stringently. Previously, some COVID-19-containment measures (e.g., social distancing) were found to reduce stress, while others (e.g., more days at home) led to isolation and increased stress [35]. These findings could account for the lack of an association between these two variables in our study.

Our model could be useful in understanding and addressing the psychological impact of possible new waves of COVID-19 cases and other epidemics. Mitigation measures boost worries and stress, particularly for those individuals who feel that these measures are not sufficient or go too far. Highlighting positive aspects about the crisis and convincing people of the effectiveness and the necessity of containment measures may not only boost compliance, but also decrease stress, since individuals feel protected by the authorities and experience less worries. This, in turn, will hopefully limit the impact on mental health as a consequence of the crisis.

Providing support is another important way to mitigate worries, enable coping, and reduce stress.

Taken together, to mitigate stress, authorities should explain containment measures well, highlight positive aspects of the crisis, address worries, and facilitate support. Since agreement with the containment measures may decrease stress, it is crucial that the measures are well-explained and their importance emphasized, and that measures are backed up by scientific evidence. Since the protection of others is an important motivating factor to restrict one's own everyday life [36], the authorities could thus emphasize that compliance with containment measures helps to protect those in need of protection. Furthermore, our analyses show that support through communities, relatives, and employers seems to be key in preventing stress; thus, these support networks should be encouraged to provide help. We identified young individuals as a group experiencing high stress levels. A recent study suggests that social media are particularly well-suited to reach out to adolescents [37]. Since we show that worries are a key driver for stress, our study also provides arguments for economic support, such as stimulus checks or short-time work. In addition, our study has identified groups that are particularly affected by the crisis. This information could help to distribute resources and target efforts.

There are some limitations to the study. First, the cross-sectional design of the study limited the ability to make inferences about the directions of causality. Second, the sample is not representative in terms of gender; as in most studies using the PSS, women are somewhat overrepresented. Third, we could not assess all individual characteristics that could have an influence on stress levels. For example, neuroticism or alexithymia could be relevant traits that make individuals particularly vulnerable to crises such as a pandemic [38]. It would be interesting to investigate the role of these characteristics in future studies.

## Supporting information

**S1 Appendix. Detailed description of materials and methods.** This appendix contains all materials and translations used in the survey, which were developed by the authors.
(PDF)

## Author Contributions

**Conceptualization:** Bartholomäus Wissmath, Fred W. Mast, Fabian Kraus.

**Data curation:** David Weibel.

**Formal analysis:** Fabian Kraus, David Weibel.

**Investigation:** Bartholomäus Wissmath, Fabian Kraus.

**Methodology:** Bartholomäus Wissmath, Fred W. Mast, David Weibel.

**Project administration:** Bartholomäus Wissmath, Fabian Kraus, David Weibel.

**Supervision:** David Weibel.

**Writing – original draft:** Bartholomäus Wissmath, Fred W. Mast, David Weibel.

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
