## [Decision Letter · Decision Letter 0]

14 Sep 2020

PONE-D-20-16793

Understanding the psychological impact of the COVID-19 pandemic and containment measures: an empirical model of stress.

PLOS ONE

Dear Dr.Wissmath,

Thank you for submitting your manuscript to PLOS ONE. After careful consideration, we feel that it has merit but does not fully meet PLOS ONE’s publication criteria as it currently stands. Therefore, we invite you to submit a revised version of the manuscript that addresses the points raised during the review process.

Please address Reviewer 3 concerns regarding:

aims and objectives

methods

discussion

Please make the changes to the abstract suggested by Reviewer One.

We look forward to receiving your revised manuscript.

Kind regards,

Rosemary Frey

Academic Editor

PLOS ONE

Journal Requirements:

Reviewers' comments:

Reviewer's Responses to Questions

**Comments to the Author**

1. Is the manuscript technically sound, and do the data support the conclusions?

Reviewer #1: Yes

Reviewer #2: Yes

Reviewer #3: Yes

2. Has the statistical analysis been performed appropriately and rigorously? 

Reviewer #1: Yes

Reviewer #2: Yes

Reviewer #3: Yes

3. Have the authors made all data underlying the findings in their manuscript fully available?

Reviewer #1: Yes

Reviewer #2: Yes

Reviewer #3: Yes

4. Is the manuscript presented in an intelligible fashion and written in standard English?

Reviewer #1: Yes

Reviewer #2: Yes

Reviewer #3: Yes

5. Review Comments to the Author

Reviewer #1: This paper was well written, with clear explanations and use of data.

The only thing I would suggest rewriting is the abstract. The conclusion is alive and hard hitting, but somehow the abstract isn't. Maybe the first sentence could be stronger; "Research suggests" is a very woolly start.

Reviewer #2: The article describes a technically-sound, well-executed study with clear explanation of research methods and sound statistical analysis. Ethical approval has been given for the study and there are no outstanding ethical issues. The data underlying the findings had been made available. The conclusions are appropriate to the data presented and findings.

Reviewer #3: This is, in summary, an interesting study conducting a path analysis to gain a deeper understanding of the psychological mechanisms during lockdown. The authors reported that experiencing fear of the disease is a key driver for being worried. They added that, based on the present model, worries about the individual, social, and economic consequences of the crisis, strongly boost stress. Importantly, the infection rate in the canton of residence also contributes to stress. Finally, positive thinking and perceived social, organizational, and governmental support mitigate worries and stress.

The authors may find as follows my main comments/suggestions.

First, when throughout the Introduction section, the authors correctly reported that COVID-19 evokes stress and that reduced social interactions are risk factors for mental disorders such as major depression, they could even mention the association between stress, depression and negative outcomes. In particular, the unique sensory processing patterns of individuals with major affective disorders and their relationship with psychiatric symptomatology and clinicial outcomes have been reported. Hyposensitivity or hypersensitivity may be "trait" markers of individuals with major affective disorders and interventions should refer to the individual unique sensory profiles and their behavioral and functional impact in the context of real life. Thus, given the above information, i suggest to cite, within the main text, the following papers about this topic (PMID: 28855878, 31103905, 31031655). In addition, subjects with a history of childhood maltreatment may be even at increased risk of negative outocomes such as suicidal behavior. Specifically, the relation between childhood maltreatment, depression, and suicidal behaviors has been demonstrated. Importantly, the exposure to abuse and neglect as a child may enhance the risk to develop both symptoms of depression and higher suicidal risk. Thus, in order to briefly discuss this topic (although i understand that the link between depression, childhood maltreatment, and suicidal behavior is not the main topic of the present manuscript), my suggestion is to cite, within the main text, the following additional manuscripts (PMID: 25169890; 32824995; 30551339).

Moreover, the most relevant aims/objectives and study hypotheses of the present study need to be better described within the main text.

Importantly, while the authors reported that they investigated a total of 1565 individuals who completed the online survey, they did not specify how many individuals were excluded to participate.

In addition, the most relevant psychometric instruments reported in the present manuscript could be described more succinctly.

Furthermore, the major shortcomings/limitations of this paper need to be better discussed as the description of the main caveats has been not included within the main text.

Finally, what is the take-home message of the present manuscript? While the authors reported that providing support is an important way to mitigate worries, enable coping, and reduce stress, they could provide some additional conclusive remarks about this topic. Specifically, how worries and stress might be really mitigated? How protective coping strategies can be implemented? Here, more details/information are needed.

6. PLOS authors have the option to publish the peer review history of their article (what does this mean?). If published, this will include your full peer review and any attached files.

Reviewer #1: **Yes: **Dr Stephen Jacobs

Reviewer #2: No

Reviewer #3: No

---

## [Author Response · Author response to Decision Letter 0]

30 Oct 2020

Reply to Reviewer 1’s comments

Reviewer 1 writes that our manuscript is “well written, with clear explanations and use of data”, with only one suggestion for improvement pertaining to the abstract. We have addressed this issue in the following reply.

Reviewer 1: “The only thing I would suggest rewriting is the abstract. The conclusion is alive and hard hitting, but somehow the abstract isn't. Maybe the first sentence could be stronger; ‘Research suggests’ is a very woolly start.”

Reply: We agree with the reviewer that the abstract could be improved and have therefore revised it. The beginning of the abstract has been modified and the conclusions restructured to give weight on the impact of our findings. The revised abstract now reads as follows:

“Epidemics such as COVID-19 and corresponding containment measures are assumed to cause psychological stress. In a survey during the lockdown in Switzerland (n=1565), we found substantially increased levels of stress in the population. In particular, individuals who did not agree with the containment measures, as well as those who saw nothing positive in the crisis, experienced high levels of stress. In contrast, individuals who are part of a risk group or who are working in healthcare or in essential shops experienced similar stress levels as the general public. The psychological mechanisms that determine stress, caused by the COVID-19 pandemic and containment measures, are not yet clear. Thus, we conducted a path analysis to gain a deeper understanding of the psychological mechanisms that lead to stress. Experiencing fear of the disease is a key driver for being worried. Our model further shows that worries about the individual, social, and economic consequences of the crisis, strongly boost stress. The infection rate in the canton (i.e. state) of residence also contributes to stress. Positive thinking and perceived social, organizational, and governmental support mitigate worries and stress. Our findings indicate that containment measures increase worries and stress, especially for those who feel that these measures are either not sufficient or go too far. Thus, highlighting positive aspects of the crisis and convincing people of the effectiveness and necessity of mitigation measures can, not only promote compliance, but also reduce stress. Our model suggests that people who feel protected by the authorities have fewer worries, which can, in turn limit the negative impact of the crisis on mental health.”

Reply to Reviewer 2’s comments

Reviewer 2 states that our study is “technically-sound, well-executed […] with clear explanation of research methods and sound statistical analysis.” Furthermore, he or she notes that there are no ethical concerns, that the data are available, and that the conclusions are appropriate. 

Reply: We appreciate Reviewer 2’s positive evaluation of our manuscript and we are grateful for his or her feedback.

Reply to Reviewer 3’s comments

Reviewer 3 concludes that our study is interesting. However, he or she raises some questions and provides suggestions for improvement. In the following section, we present Reviewer 3's recommendations to improve the paper and address each point in detail.

Reviewer 3: When throughout the Introduction section, the authors correctly reported that COVID-19 evokes stress and that reduced social interactions are risk factors for mental disorders such as major depression, they could even mention the association between stress, depression and negative outcomes. In particular, the unique sensory processing patterns of individuals with major affective disorders and their relationship with psychiatric symptomatology and clinicial outcomes have been reported. Hyposensitivity or hypersensitivity may be "trait" markers of individuals with major affective disorders and interventions should refer to the individual unique sensory profiles and their behavioral and functional impact in the context of real life. Thus, given the above information, I suggest to cite, within the main text, the following papers about this topic (PMID: 28855878, 31103905, 31031655). In addition, subjects with a history of childhood maltreatment may be even at increased risk of negative outcomes such as suicidal behavior. Specifically, the relation between childhood maltreatment, depression, and suicidal behaviors has been demonstrated. Importantly, the exposure to abuse and neglect as a child may enhance the risk to develop both symptoms of depression and higher suicidal risk. Thus, in order to briefly discuss this topic (although I understand that the link between depression, childhood maltreatment, and suicidal behavior is not the main topic of the present manuscript), my suggestion is to cite, within the main text, the following additional manuscripts (PMID: 25169890; 32824995; 30551339).

Reply: We thank the reviewer for these suggestions. We write in our paper that certain individuals are particularly vulnerable to consequences of the pandemic and the containment measures. We agree with the reviewer that existing research shows that alexithymia, childhood maltreatment, as well as neuroticism, can be such risk factors. In the revised version, we mention this: 

“Some individuals may be more vulnerable to psychosocial consequences; risk factors include certain personality traits (e.g. neuroticism, alexithymia), mental or physical pre-existing conditions, as well as previous traumatic experiences (e.g. childhood maltreatment), or the individual’s current situation, such as working in healthcare, social isolation, or poverty [4-10]” (see manuscript without track changes, p. 3, line 39ff)

We agree with the reviewer that a comprehensive exploration of the impact of childhood maltreatment and alexithymia es somewhat beyond the scope of our study, but we find the inputs useful and have incorporated them later in the discussion. Our study shows that individuals who do not see anything positive about the pandemic show strongly increased stress levels. The question arises as to how this can be explained. Since childhood maltreatment is associated with hopelessness and lack of perspective, it is possible that the experience of maltreatment in the past could play a role. We have incorporated this into the conclusion, the corresponding passage reads as follows: 

“Although seeing something positive about the crisis may be associated with the individual situation, personality traits, and the respondent’s coping behavior [17], our findings indicate that recognizing positive aspects could alleviate stress. Some individuals, however, do not see anything positive about the crisis. This raises the question as to why their assessment of the situation is so negative. For example, childhood maltreatment is associated with hopelessness [20] and psychological distress [5]. A recent study in connection with COVID-19 indicates that pre-pandemic maltreatment makes people particularly vulnerable to negative psychosocial consequences of the pandemic [21]. In addition, other factors and traits – such as pessimistic thinking [e.g. 22] or generalized anxiety [e.g. 23] – could account for the lack of perspective.” (see manuscript without track changes, p. 7, line 110ff) 

Furthermore, it is now mentioned in the manuscript (limitation section) that it would be promising to examine the role of alexithymia in further studies (see also reply to answer to point 4). 

Additional references in the revised version (in the same order as they appear in the article):

Liu S, Lithopoulos A, Zhang CQ, Garcia-Barrera MA, Rhodes RE. Personality and perceived stress during COVID-19 pandemic: Testing the mediating role of perceived threat and efficacy. Pers Individ Dif. 2021;168:110351. Epub 2020 Aug 21. 

De Berardis D, Fornaro M, Orsolini L, Valchera A, Carano A, Vellante F et al. Alexithymia and suicide risk in psychiatric disorders: a mini-review. Front Psychiatry. 2017;8:148. 

Pompili M, Innamorati M, Lamis DA, Erbuto D, Venturini P, Ricci F, Serafini G, Amore M, Girardi P. The associations among childhood maltreatment, "male depression" and suicide risk in psychiatric patients. Psychiatry Res. 2014;220(1-2):571-8. 

Santiago, C. D., Wadsworth, M. E., & Stump, J. (2011). Socioeconomic status, neighborhood disadvantage, and poverty-related stress: Prospective effects on psychological syndromes among diverse low-income families. Journal of Economic Psychology, 32(2), 218-230.

Wanklyn SG, Day DM, Hart TA, Girard TA. Cumulative childhood maltreatment and depression among incarcerated youth: impulsivity and hopelessness as potential intervening variables. Child Maltreat. 2012;17(4):306-17.

Guo J, Fu M, Liu D, Zhang B, Wang X, van IJzendoorn MH. Is the psychological impact of exposure to COVID-19 stronger in adolescents with pre-pandemic maltreatment experiences? A survey of rural Chinese adolescents. Child Abuse Negl. 2020;20:104667.

Norem JK. Defensive pessimism, optimism, and pessimism. In: Chang EC, editor, Optimism & pessimism: Implications for theory, research, and practice. Washington DC: American Psychological Association; 2001. p. 77–100.

Miranda R, Mennin DS. Depression, generalized anxiety disorder, and certainty in pessimistic predictions about the future. Cognit Ther Res. 2007;31(1),71-82.

Betsch, C. How behavioural science data helps mitigate the COVID-19 crisis. Nat Hum Behav. 2020;4:438.

Andrews JL, Foulkes L, Blakemore SJ. Peer Influence in Adolescence: Public-Health Implications for COVID-19. Trends Cogn Sci. 2020;24(8):585-587.

Tang W, Hu T, Yang L, Xu J. The role of alexithymia in the mental health problems of home-quarantined university students during the COVID-19 pandemic in China. Pers Individ Dif. 2020;165:110131.

Reviewer 3: The most relevant aims/objectives and study hypotheses of the present study need to be better described within the main text.

Reply: We agree with the reviewer that the aim of the study could be better explained. We now describe this in more detail in the revised version. Here is the relevant text passage:

“However, little is known about the psychological mechanisms that determine individual stress in this crisis. It is still unclear which individuals are particularly affected by stress, which factors have an increasing or decreasing influence on stress, and how these factors interact. In particular, the interplay between fear of the virus, infection rate, risk factors, individual and collective resources, perception of the containment strategy, and psychological stress has not yet been fully understood. The aim of the present study is to address these questions by using path analysis.” (see manuscript without track changes, p. 3, line 52ff)

Reviewer 3: While the authors reported that they investigated a total of 1565 individuals who completed the online survey, they did not specify how many individuals were excluded to participate. In addition, the most relevant psychometric instruments reported in the present manuscript could be described more succinctly.

Reply: The number of participants refers to those who completed the survey and did not request the deletion of their data. 215 individuals had not completed or had cancelled the survey. The data of three participants were removed since they had requested the deletion of their data after completion of the survey. This is now described in the manuscript. We reassessed the description of the psychometric instruments and made it more succinct wherever possible. However, we would like to mention that the editor (see below) has asked us to provide sufficient information regarding the survey and questionnaire items.

Reviewer 3: The major shortcomings/limitations of this paper need to be better discussed as the description of the main caveats has been not included within the main text.

Reply: We thank the reviewer for bringing this point to our attention. We have now added a passage at the end of the Conclusion section, which addresses limitations and possible future studies. We also mention that it would be useful to examine the role of alexithymia and neuroticism in more detail. The additional passage reads as follows:

“There are some limitations to the study. First, the cross-sectional design of the study limited the ability to make inferences about the directions of causality. Second, the sample is not representative in terms of gender; as in most studies using the PSS, women are somewhat overrepresented. Third, we could not assess all individual characteristics that could have an influence on stress levels. For example, neuroticism or alexithymia could be relevant traits that make individuals particularly vulnerable to crises such as a pandemic [26]. It would be interesting to investigate the role of these characteristics in future studies.” (see manuscript without track changes, p. 9, line 145ff)

Reviewer 3: What is the take-home message of the present manuscript? While the authors reported that providing support is an important way to mitigate worries, enable coping, and reduce stress, they could provide some additional conclusive remarks about this topic. Specifically, how worries and stress might be really mitigated? How protective coping strategies can be implemented? Here, more details/information are needed.

Reply: The take-home message is that the pandemic significantly increases stress levels and that this increase depends on a number of factors, with support, communicating positive aspects, and advocacy of the authorities' measures as being particularly important. We believe that we have already clearly stated this in the manuscript. However, we agree with the reviewer that we could conclude the discussion with additional suggestions on what could be done to mitigate stress. We have included a corresponding passage in the revised manuscript:

“Taken together, to mitigate stress, authorities should explain containment measures well, highlight positive aspects of the crisis, address worries, and facilitate support. Since agreement with the containment measures may decrease stress, it is crucial that the measures are well-explained and their importance emphasized, and that measures are backed up by scientific evidence. Since the protection of others is an important motivating factor to restrict one's own everyday life [24], the authorities could thus emphasize that compliance with containment measures helps to protect those in need of protection. Furthermore, our analyses show that support through communities, relatives, and employers seems to be key in preventing stress; thus, these support networks should be encouraged to provide help. We identified young individuals as a group experiencing high stress levels. A recent study suggests that social media are particularly well-suited to reach out to adolescents [25]. Since we show that worries are a key driver for stress, our study also provides arguments for economic support, such as stimulus checks or short-time work. In addition, our study has identified groups that are particularly affected by the crisis. This information could help to distribute resources and target efforts.” (see manuscript without track changes, p. 8, line 131ff)

Reply to additional comments from the Editor

The editorial office further mentions two additional requirements that need to be addressed. 

Editor: Please ensure that your manuscript meets PLOS ONE's style requirements, including those for file naming. 

Reply: As requested, it is ensured that all formal requirements are met.

Editor: Please include additional information regarding the survey or questionnaire used in the study and ensure that you have provided sufficient details that others could replicate the analyses. For instance, if you developed a questionnaire as part of this study and it is not under a copyright more restrictive than CC-BY, please include a copy, in both the original language and English, as Supporting Information.

Reply: As requested, we have added some additional information in order to provide sufficient details (see appendix S1).

---

## [Decision Letter · Decision Letter 1]

19 May 2021

PONE-D-20-16793R1

Understanding the psychological impact of the COVID-19 pandemic and containment measures: an empirical model of stress.

PLOS ONE

Dear Dr. Wissmath,

Thank you for submitting your manuscript to PLOS ONE. After careful consideration, we feel that it has merit but does not fully meet PLOS ONE’s publication criteria as it currently stands. Therefore, we invite you to submit a revised version of the manuscript that addresses the points raised during the review process.

Thank you for your resubmission.  While this version is improved, methodological issues have been raised by reviewer 4 that must be addressed before publication is considered.

We look forward to receiving your revised manuscript.

Kind regards,

Rosemary Frey

Academic Editor

PLOS ONE

Journal Requirements:

Reviewers' comments:

Reviewer's Responses to Questions

**Comments to the Author**

1. If the authors have adequately addressed your comments raised in a previous round of review and you feel that this manuscript is now acceptable for publication, you may indicate that here to bypass the “Comments to the Author” section, enter your conflict of interest statement in the “Confidential to Editor” section, and submit your "Accept" recommendation.

Reviewer #3: All comments have been addressed

Reviewer #4: (No Response)

2. Is the manuscript technically sound, and do the data support the conclusions?

Reviewer #3: Yes

Reviewer #4: (No Response)

3. Has the statistical analysis been performed appropriately and rigorously? 

Reviewer #3: (No Response)

Reviewer #4: (No Response)

4. Have the authors made all data underlying the findings in their manuscript fully available?

Reviewer #3: Yes

Reviewer #4: (No Response)

5. Is the manuscript presented in an intelligible fashion and written in standard English?

Reviewer #3: Yes

Reviewer #4: (No Response)

6. Review Comments to the Author

Reviewer #3: In the revised manuscript, the authors addressed most of the major questions raised by Reviewers improving both the main structure and quality of the present paper. I have no further additional comments.

Reviewer #4: - Add a section of Statistical Analysis: include information related to statistical analysis, type of variable and data presentation, processing , program used for data analysis and version and manufacturing company.

what program was used for creation of path model, what test of significance were used , did you test for normality , and what was the selected level of significance.

I suggest adding more information regard path model , what was considered significance level for paths, it seems you choose different statistical level for the paths and the tests in table 1 (0.01 vs 0.001)elaborate more, based on what certain variable were included in the model while other were not?

- Follow the sequence of Introduction, Methods. Results and Discussion so the methods section has to be before the result.

The above 2 points are major and has to be added/corrected.

the rest are minor modifications

- Comment for different part of manuscript are available at attached pdf file in the comment section (less than 29 point) most of them are simple modifications, kindly correct them.

- Note that the methodology section has to be clear enough so if a different investigator conducted similar study, he will get comparable result. so read it again and modify.

- The question 'Following the shelter in place-orders stringently'

There is no mention for this question elsewhere in methodology and even was not discussed in the discussion section. I think it's important to have few word on this question in methodology and discussion as following shelter in order is different from just agreement with containment strategy.

- Certain points regard the approaching of the study sample has to be clarified like what platform were used and how the author make sure that the respondents will not fill up the questionnaire twice? how did they know if they were Swiss residents? as there is no mention of specific question or info indicating that thing while surveying the study group.

- Proof reading has to be done as a ref. error are found in the discussion section and some sentences need to be modified

Thank you for your contribution to this area of interest.

7. PLOS authors have the option to publish the peer review history of their article (what does this mean?). If published, this will include your full peer review and any attached files.

Reviewer #3: No

Reviewer #4: No

---

## [Author Response · Author response to Decision Letter 1]

17 Jun 2021

Reply to Reviewer 3’s comments

According to Reviewer 3, all comments have been addressed. He states: “In the revised manuscript, the authors addressed most of the major questions raised by Reviewers improving both the main structure and quality of the present paper. I have no further additional comments.“

Reply: We appreciate Reviewer 3’s positive evaluation and we are thankful for his or her positive feedback.

Reply to Reviewer 4’s comments

Reviewer 4 raises some questions and concerns and provides suggestions for improvement. In the following paragraphs, we present Reviewer 4's recommendations to improve the paper and address each point in detail.

Comment 1:

Add a section of Statistical Analysis: include information related to statistical analysis, type of variable and data presentation, processing, program used for data analysis and version and manufacturing company.

what program was used for creation of path model, what test of significance were used, did you test for normality, and what was the selected level of significance.

I suggest adding more information regard path model, what was considered significance level for paths, it seems you choose different statistical level for the paths and the tests in table 1 (0.01 vs 0.001) elaborate more, based on what certain variable were included in the model while other were not?

Reply:

We thank the reviewer for bringing these aspects to our attention and agree that further information is needed. We have now added a new section, “Statistical analysis,” presenting the details of the statistical analysis and the software used to conduct the analysis. Furthermore, we clarified our choice of variables for the path model, and supplemented and corrected Table 1 with additional information about levels of significance. As suggested, we additionally added a percentages column to Table 1. The section reads as follows:

“In a first step, a one-sample z-test was conducted to compare the stress level of the study sample to a representative community sample under normal conditions [22]. Before the test was carried out, we checked whether the stress values were normally distributed: the stress scores were approximately normally distributed, as the skewness equaled 0.43 (a normal distribution can be assumed for skewness values between -0.5 and 0.5 [24]). Thus, the data allowed for z-test as well as ANOVAs (step 2 below).

In a second step, an ANOVA was carried out to compare different subgroups of the study sample. Because of multiple comparisons, Bonferroni's adjustments were made to prevent Type I error inflation (α = 0.05/14 = 0.00357). IBM SPSS Statistics for Windows, Version 25.0 was used to run this analysis [25].

In a third step, we tested how stress is related to various variables that are assumed to affect well-being in the context of the pandemic. A path analysis was conducted with stress as the target variable. The path analysis was carried out with IBM SPSS Amos, Version 25.0 [26], using the maximum likelihood method. To test for normality prior to the analysis, a descriptive approach was used. In the context of SEM or path analysis, kurtosis values of > 7 indicate a substantial deviation from normality [27, 28]. In terms of skewness values of > 3 indicate extreme levels of skewness [29]. For both skewness and kurtosis, our values were below these thresholds and thus, the use of the maximum likelihood method was appropriate. To test the significance of the path coefficients, t-tests were calculated. The following variables were included in the path model: stress, fear of COVID-19, worries, support, and infection rate. These variables were selected because previous findings suggest that they affect stress. In addition, "agreement with containment measures" as well as "positive aspects" were included, because the second step of analysis (see above) showed that these variables exert the strongest influence on stress. 

We used p < 0.05 as a priori level of significance. However, we reported lower p-values when appropriate (e.g., p < 0.001). Effect sizes were reported as small, medium, or strong according to the heuristics of Eid, Gollwitzer and Schmitt [30].” (Lines 125-150)

Comment 2:

Follow the sequence of Introduction, Methods. Results and Discussion so the methods section has to be before the result.

Reply:

We agree that the requested order of sections makes it easier for the reader to understand the results section and therefore improves the readability of the manuscript. As suggested, we have changed the order of the sections, which now follows the requested order: Introduction – Materials and methods – Results – Discussion. 

Comment 3 and comment 4:

Comment for different part of manuscript are available at attached pdf file in the comment section (less than 29 point) most of them are simple modifications, kindly correct them.

Note that the methodology section has to be clear enough so if a different investigator conducted similar study, he will get comparable result. so read it again and modify.

Reply:

We thank the reviewer for the thorough and detailed corrections of these aspects. We have corrected details as requested and added further information where necessary. 

Comment 5

The question 'Following the shelter in place-orders stringently'

There is no mention for this question elsewhere in methodology and even was not discussed in the discussion section. I think it's important to have few word on this question in methodology and discussion as following shelter in order is different from just agreement with containment strategy.

Reply:

The description of this question has now been added to the materials and methods section (see line 114 in the revised manuscript). Although there were no significant associations between this variable and any other target variable, we agree with the reviewer that this variable differs from agreement with containment strategy and therefore deserves to be discussed. We added a new paragraph about this variable in the discussion section. This reads as follows:

“In contrast to agreement with the government’s containment strategies, there was no association between following the shelter-in-place orders stringently and perceived stress. About a third of the respondents had indicated that they did not follow the orders stringently. Previously, some COVID-19-containment measures (e.g., social distancing) were found to reduce stress, while others (e.g., more days at home) led to isolation and increased stress [35]. These findings could account for the lack of an association between these two variables in our study.” (Lines 214 - 220)

Comment 6

Certain points regard the approaching of the study sample has to be clarified like what platform were used and how the author make sure that the respondents will not fill up the questionnaire twice? how did they know if they were Swiss residents? as there is no mention of specific question or info indicating that thing while surveying the study group.

Reply:

As suggested, we have added further information about the study sample and its recruitment. Concerning the possibility of participants filling out the survey twice, we would like to point out that for privacy reasons, as little data as possible was collected about the participants. For example, no IP addresses were stored. Thus, it cannot be ruled out that people took part in the survey more than once. An examination of the data for repeated or non-human responses did not reveal any indication of such responses though. The newly added section reads as follows:

“An examination of the data for repeated or non-human responses did not reveal any indication of such responses. According to their demographic information, all participants resided in a Swiss canton (as opposed to living abroad). Further information about the sample is provided in Table 1.” (Lines 74 - 77)

Comment 7

Proof reading has to be done as a ref. error are found in the discussion section and some sentences need to be modified

Reply:

MS was proofread by native speakers again. We have revised the manuscript accordingly.

---

## [Decision Letter · Decision Letter 2]

7 Jul 2021

Understanding the psychological impact of the COVID-19 pandemic and containment measures: an empirical model of stress.

PONE-D-20-16793R2

Dear Dr..Wissmath,

We’re pleased to inform you that your manuscript has been judged scientifically suitable for publication and will be formally accepted for publication once it meets all outstanding technical requirements.

Kind regards,

Rosemary Frey

Academic Editor

PLOS ONE

Additional Editor Comments (optional):

Reviewers' comments:

Reviewer's Responses to Questions

**Comments to the Author**

1. If the authors have adequately addressed your comments raised in a previous round of review and you feel that this manuscript is now acceptable for publication, you may indicate that here to bypass the “Comments to the Author” section, enter your conflict of interest statement in the “Confidential to Editor” section, and submit your "Accept" recommendation.

Reviewer #3: All comments have been addressed

Reviewer #4: (No Response)

2. Is the manuscript technically sound, and do the data support the conclusions?

Reviewer #3: Yes

Reviewer #4: (No Response)

3. Has the statistical analysis been performed appropriately and rigorously? 

Reviewer #3: Yes

Reviewer #4: (No Response)

4. Have the authors made all data underlying the findings in their manuscript fully available?

Reviewer #3: Yes

Reviewer #4: (No Response)

5. Is the manuscript presented in an intelligible fashion and written in standard English?

Reviewer #3: Yes

Reviewer #4: (No Response)

6. Review Comments to the Author

Reviewer #3: In the revised manuscript, the authors addressed successfully most of the Reviewers' comments. I have no further additional comments.

Reviewer #4: Thank you for the effort you put into this MS

please correct the line 210 in discussion

((In addition, other factors and traits –

210 such as pessimistic thinking [e.g., 33] or generalized anxiety [e.g., 34]))

the 33 and 34 appear as references so remove the (e.g)

Regards

7. PLOS authors have the option to publish the peer review history of their article (what does this mean?). If published, this will include your full peer review and any attached files.

Reviewer #3: No

Reviewer #4: No

---

## [Editor Report · Acceptance letter]

12 Jul 2021

PONE-D-20-16793R2 

Understanding the psychological impact of the COVID-19 pandemic and containment measures: an empirical model of stress 

Dear Dr. Wissmath:

I'm pleased to inform you that your manuscript has been deemed suitable for publication in PLOS ONE. Congratulations! Your manuscript is now with our production department. 

Kind regards, 

on behalf of

Dr. Rosemary Frey 

Academic Editor

PLOS ONE